# Exposure assessment of adults living near unconventional oil and natural gas development and reported health symptoms in southwest Pennsylvania, USA

**Hannah N. Blinn** [1,2]☯*, **Ryan M. Utz**[1]☯, **Lydia H. Greiner**[2‡], **David R. Brown**[2‡]

**1** Falk School of Sustainability, Chatham University, Gibsonia, Pennsylvania, United States of America,
**2** Southwest Pennsylvania Environmental Health Project, McMurray, Pennsylvania, United States of America

☯ These authors contributed equally to this work.
‡ These authors also contributed equally to this work.
\* hnblinn@gmail.com

**Data Availability Statement:** Gas well location and emissions data is hosted on a PowerBI report and controlled by the PA Department of Environmental Protection. To view only gas well data, filter by

## Abstract

Recent research has shown relationships between health outcomes and residence proximity to unconventional oil and natural gas development (UOGD). The challenge of connecting health outcomes to environmental stressors requires ongoing research with new methodological approaches. We investigated UOGD density and well emissions and their association with symptom reporting by residents of southwest Pennsylvania. A retrospective analysis was conducted on 104 unique, de-identified health assessments completed from 2012–2017 by residents living in proximity to UOGD. A novel approach to comparing estimates of exposure was taken. Generalized linear modeling was used to ascertain the relationship between symptom counts and estimated UOGD exposure, while Threshold Indicator Taxa Analysis (TITAN) was used to identify associations between individual symptoms and estimated UOGD exposure. We used three estimates of exposure: cumulative well density (CWD), inverse distance weighting (IDW) of wells, and annual emission concentrations (AEC) from wells within 5 km of respondents' homes. Taking well emissions reported to the Pennsylvania Department of Environmental Protection, an air dispersion and screening model was used to estimate an emissions concentration at residences. When controlling for age, sex, and smoker status, each exposure estimate predicted total number of reported symptoms (CWD, $p<0.001$; IDW, $p<0.001$; AEC, $p<0.05$). Akaike information criterion values revealed that CWD was the better predictor of adverse health symptoms in our sample. Two groups of symptoms (i.e., eyes, ears, nose, throat; neurological and muscular) constituted 50% of reported symptoms across exposures, suggesting these groupings of symptoms may be more likely reported by respondents when UOGD intensity increases. Our results do not confirm that UOGD was the direct cause of the reported symptoms but raise concern about the growing number of wells around residential areas. Our approach presents a novel method of quantifying exposures and relating them to reported health symptoms.

Facility Type. We additionally filtered by year, county, and and pollutant as described in our methods. Data can then be exported to a .csv file: http://www.depgreenport.state.pa.us/ powerbiproxy/powerbi/Public/DEP/AQ/PBI/Air_ Emissions_Report Climate data was retrieved from NOAA's local climatological database. To use the tool, you need to select the state and county of where the airport is located. We used data from the Pittsburgh Allegheny County Airport in Allegheny County, PA. Once the airport has been added to your cart, you can determine the data range you wish to download and request a .csv of the data: https://www.ncdc.noaa.gov/cdo-web/datatools/lcd Health data cannot be shared publicly because some of the data we collect is in rural areas with sparse population. In areas of sparse population, it may be possible to identify participants using data such as GIS coding. Data are available from the Environmental Health Project Institutional Data Access / Ethics Committee (contact via Environmental Health Project, Sarah Rankin 724.260.5504) for researchers who meet the criteria for access to confidential data.

**Funding:** DB and LG positions at the Southwest PA Environmental Health Proejct are funded by the Heinz Endowments E5450. The funders did not play a role in this study's design analysis, decision to publish, or preparation of the manuscript. Their funding was used prior to this study when the data was being collected. This study is a retrospective review of that data. HB and RU did not receive funding for this project.

**Competing interests:** The authors have declared that no competing interests exist.

## Introduction

Unconventional oil and natural gas development (UOGD) may represent a health risk due to exposure to chemicals used during the hydraulic fracturing process, on-site emissions, and/or a lack of strict regulations [1–4]. The UOGD process involves a combination of horizontal drilling across shale formations and the use of a heterogeneous fracturing fluid injected into wells at high pressure to fracture shale and release trapped oil and gas. Evidence suggesting associations between UOGD activity and adverse health effects has emerged from multiple studies. UOGD activity has been associated with adverse birth outcomes [5–7], increased rates of hospital use [8–10], asthma [11,12], and upper respiratory and neurologic symptoms [13–15]. These studies have used a variety of approaches to estimate exposure to UOGD, including inverse distance weighting (IDW), cumulative well count, cumulative well density (CWD), well activity metrics, spatiotemporal models, and direct water sampling [6–8,13,16,17].

Given the associations between UOGD development and adverse health outcomes, but lack of resolution on questions pertaining to safe proximity of residency to wells, we sought to determine which variables related to UOGD are associated with a higher number of reported symptoms. For this study, two proximity metrics and one exposure variable constitute our exposure estimates and are referred to as exposure measures throughout this paper. This study was conducted to address the following questions: 1) Which exposure measure(s) best predicts the of number of symptoms reported? and 2) Which individual symptoms are associated with increasing exposure as estimated by each exposure measure? Unlike prior studies, this analysis compares three estimates of exposure: CWD, an IDW measure, and annual emission concentrations (AEC) derived from estimated well emissions within 5 km of a residence. CWD is defined as the count of wells divided by a spatial scale in $km^2$ [8], while IDW, a similar measure, weights wells according to distance from a residence [6,7]. The AEC measure used publicly available data on wells to estimate concentrations of emission pollution at a residence. Bamber at al. [18] notes that exposure to UOGD is poorly characterized, and this analysis– comparing three estimates of exposure–attempts to address this concern. Though frequently used proximity and density metrics are included in this analysis, the methodological approach taken here has not been used to model emission concentrations at the home nor to predict symptom outcomes associated with increasing levels of exposure. The use of two methodologies applied here (i.e., statistical modeling to analyze the influence of different exposures on symptom reporting, and a technique to identify specific symptoms that might be indicative of exposure) suggests new techniques for studying relationships between health and exposure.

## Materials and methods

### Study sites & health outcomes

The Southwest Pennsylvania Environmental Health Project (hereafter referred to as EHP) is a nonprofit public health organization in Washington County, Pennsylvania (PA). Between February 1, 2012 and December 31, 2017, 135 children and adults completed health assessments at EHP. Individuals self-selected and approached EHP because of their concerns about exposure to UOGD. Health data were abstracted as described in Weinberger et al. [19] and the same data were used in this analysis.

As described by Weinberger et al. [19] the 135 de-identified health assessments were reviewed retrospectively by a team of health-care providers, including a board-certified occupational-health physician and at least one nurse practitioner. Records were excluded if the respondent was under 18 years old, worked in the oil-and-gas industry, lived outside of PA, or did not fully complete the assessment form (17 excluded). The remaining 118 health

assessments were reviewed. Each symptom recorded in the assessment was reviewed and those symptoms that could be plausibly explained by co-occurring medical conditions, medical history, or work and/or social history were excluded. For this analysis, symptoms that remained were grouped into nine categories: general; lung and heart; skin; eyes, ears, nose, and throat (EENT); gastrointestinal (GI); nerves and muscle; reproductive; blood system; and psychological. For this analysis, we restricted the sample to residents of southwest PA with known latitude and longitude data for their residence (14 individuals excluded). The study population included individuals from eight counties: Washington, Greene, Beaver, Butler, Allegheny, Bedford, Fayette, and Westmoreland (Fig 1). This resulted in a convenience sample of 104 adults. This study was approved by the New England Institutional Review Board and the Chatham University Institutional Review Board.

## Exposure measures

**Cumulative well density and inverse distance weighting.**   Home address was collected at the time of the health assessment. For this analysis, the address was used to determine the latitude and longitude coordinate of the residence of each respondent [21].

The PA Department of Environmental Protection (PA DEP) publishes active well locations and reported emissions on an open-access online portal [22]. The emissions inventory provides well location data in latitude and longitude coordinates and emissions data by pollutant

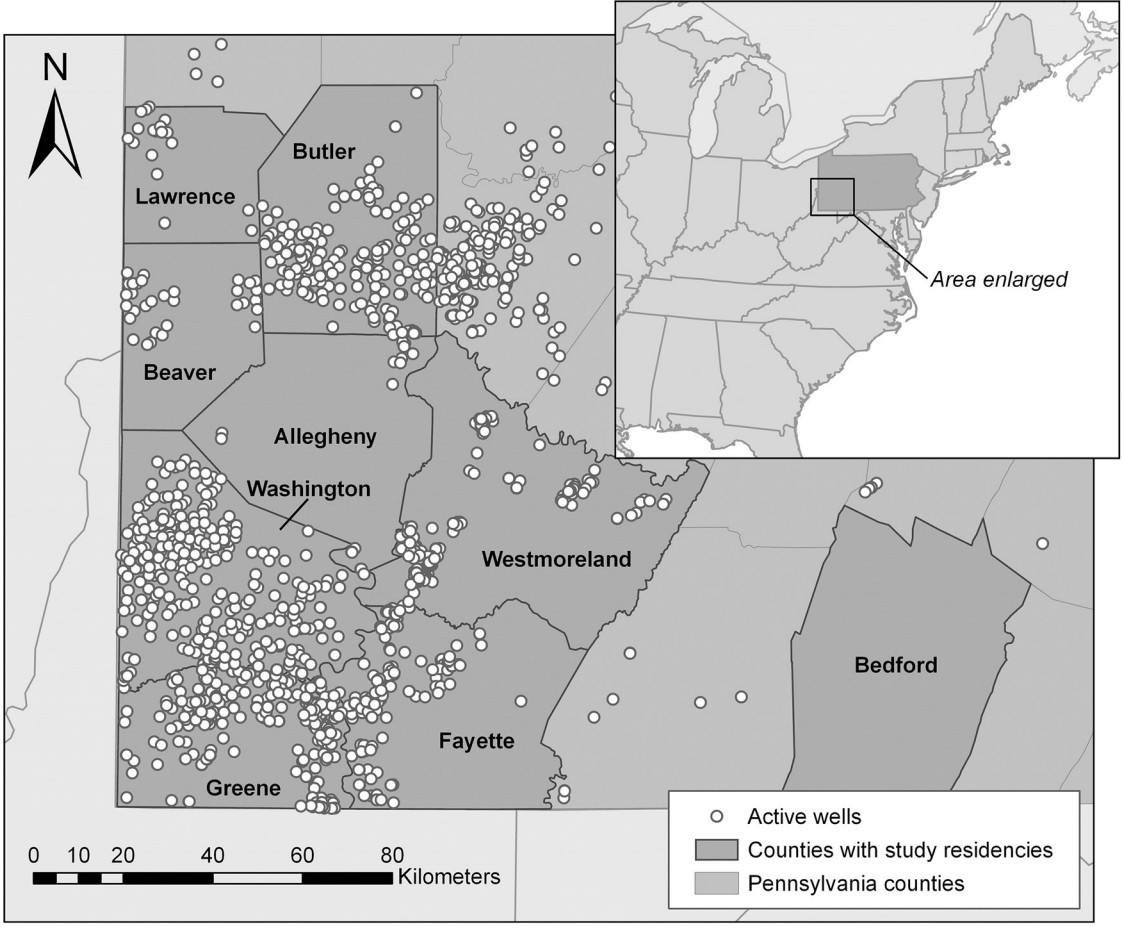

**Fig 1. Study area and active well locations.** Southwestern PA study location and active wells in 2016. No respondents lived in Lawrence County; however, a respondent in Butler County lived near the county border. Map was made with ArcGIS Desktop [20].

type for each well. For assessments completed between February 1, 2012 and December 31, 2017, ArcGIS ArcMap 10.3 [20] was used to plot the latitude and longitude of each respondent's residence alongside all active, unconventional wells within a 5-km radius around the residence during that year. A CWD was calculated for each respondent by dividing the number of wells in a 5-km radius around the home by the area of the radius.

An IDW calculation was also applied as a second method for quantifying exposure intensity. This measure applies more weight to wells located closer to a residence than to those located farther away. The inverse distance of each well within a 5-km radius of a residence was calculated, and those values were summed into one IDW score per residence as shown in the following equation:

$$IDW\ density = \sum_{i=1}^{n} \frac{1}{d_i} \tag{1}$$

where distance ($d$) is kilometers between the well ($i$) and respondent's residence, and $n$ is the number of wells within the 5-km radius [5,13]. For this analysis, only wells located within PA state lines were included in the calculations due to a lack of data availability from neighboring states. Four residences' 5-km radius crossed into neighboring West Virginia. For these sites, the radius percentages outside of Pennsylvania were 0.6%, 4.4%, 10.7%, and 14.3%.

**Annual emissions concentration.**   Annual emissions inventories for 2012 through 2017 were exported from the PA DEP's database. Sources reported on the emissions inventory included venting and blowdown, dehydration units, drill rigs, stationary engines, pneumatic pumps, fugitive emissions, and emissions produced during the well completion stage. Sources of emissions that are not represented in the inventory include flaring, off-gassing from contaminated water, and truck traffic. A review of the PA DEP's emissions-inventory data revealed six compounds had the highest reported volume expressed in tons/year: carbon monoxide, nitrogen oxides, particulate matter ($PM_{2.5}$), aggregated volatile organic compounds (VOCs), methane, and carbon dioxide [22]. To estimate emissions at the residence, we used carbon monoxide, nitrogen oxides, $PM_{2.5}$, and VOCs because they had known health effects at the expected level of exposure; methane and carbon dioxide did not so were not included despite being two of the top six compounds emitted. For this study, tons/year was converted to grams/hour.

A complete explanation of how concentrations at a residence were estimated can be found in Brown et al. [23] and will briefly be described here. To estimate emissions concentration at a respondent's residence, an atmospheric dispersion box model was used to determine air dilution downwind from emission sources (wells) and estimate the concentration of compounds at a residence. The model assumes a theoretical box, or volume, of air carries emissions downwind from a well. As the box moves away from the source, the size of the box increases, and the concentration of pollutants is proportionally diluted. The initial concentration is inversely proportional to the rate of speed with which the box moves over the source. The vertical and lateral expansion of the box as it moves downwind is determined by weather and wind speed. This screening model estimates the level of air dilution during dispersion using three parameters: 1) cloud cover, 2) wind speed, and 3) time of day. These parameters are taken from Pasquill [24]. His report identifies six stability classes and five wind speeds that characterize the meteorological conditions that define these classes [25,26]. Using these conditions, we applied hourly cloud cover and wind speed data retrieved from the National Oceanic and Atmospheric Administration (NOAA) for the years 2012 through 2017. To ensure a complete set of weather data for each year of the study, we chose to use data from one major airport in southwest PA, the Pittsburgh Allegheny County Airport in West Mifflin, PA, in the model [27]. We were able to establish hourly conditions over a year and apply the estimates to each residence in our

sample, to determine an annual level of exposure for each residence. Estimates of annual average exposures were based on weather patterns for each year over the entire region.

After our screening model was established, we used the weather data to calculate hourly concentrations from a reference well, estimated to emit 300 grams of a compound per hour, to standardize the formula when calculating how other wells deviate from a given reference [23]. Once hourly concentrations were computed for the reference case, we calculated a 90th percentile emissions concentration value ($\mu g/m^3$) for distances of 0.5 km, 1 km, 2 km, 3 km, and 5 km in the four directional quadrants around the reference well. The resulting values represent varying exposure levels experienced at a given residence living between 0.5–5 km from the reference well. The hourly emissions are assumed proportional to the 300 grams/hour reference. Using the PA DEP data for the year corresponding to the respondent's health assessment, the emissions of carbon monoxide, nitrogen oxides, $PM_{2.5}$, and VOCs in grams/hour were summed into one total for each well.

Well sites are ubiquitous around residences in these counties, so we used the model to first calculate a residence's exposure for the four directional quadrants. Within a quadrant, the distance of each well from the residence was determined and, depending on the distance, the 90th percentile concentration value was assigned to that well. Then, the total emissions from the well, in grams/hour, was multiplied by the 90th percentile concentration value and divided by 300 grams/hour to derive the deviance from the reference in each quadrant. The outputs give $\mu g/m^3$ per well for each directional quadrant in a 5-km radius. The estimated emission concentrations from each well, across all quadrants, were added together into an annual total exposure value per residence. The total exposure value was used as the AEC measure in the analysis.

## Statistical analysis

All statistical analyses were executed in the R Project for Statistical Computing [28]. Model comparisons were made using glmutli version 1.0.7.1 [29], and TITAN analyses with TITAN2 version 2.1 [30].

The analysis consisted of two approaches to address the research questions: generalized linear models (GLMs) to test the association between the number of symptoms reported and the intensity of each exposure, and Threshold Indicator Taxa Analysis (TITAN) to predict which specific symptoms were most likely to be reported with increasing intensity of each exposure measure. Each individual symptom reported in the health assessment was binomially coded per respondent with 1/0 for yes/no. An alpha level of $< = 0.05$ was used as a threshold for significance in both tests.

Because the dependent variable followed a Poisson distribution, GLMs were used for modeling. For each exposure GLM, a tool was used to automate statistical model selection by generating all possible unique combinations of our demographic variables with each exposure measure to identify the best-fit statistical model for each exposure measure against total number of symptoms. Our demographic variables included: age, sex, smoking status, and water source. All demographic variables were included in the selection tool and, by default, 100 potential models were generated *a priori* to determine the best fitting models. To choose our model, Akaike information criterion (AIC) values, with a correction for small sample sizes, and number of terms for each output model were compared [31]. Lower AIC values are associated with simpler models that exclude irrelevant terms, so when comparing models, the model with the lowest AIC is considered optimal [32,33]. The best model is the one with the lowest or second-lowest AIC score and then statistically assessed for each exposure variable [34]. Interactions between variables were excluded from the best model to increase model parsimony

and only explore main effects. Zero-inflation was not required for our data as only 15% of the sample reported no symptoms. To determine our radius distance around the home, we applied GLM analyses using three spatial scales of cumulative well density: 1, 2, and 5 km. AIC criterion was used to determine which scale to study.

To assess how individual symptoms were related to changing density (CWD and IDW) and AEC, we applied the TITAN methodology. TITAN is a non-parametric analysis traditionally applied in the ecological sciences, but increasingly applied in environmental science [35], where the presence/absence of a species (also referred to as taxon) among different samples of communities is used to assess nonlinear community-scale responses, both positive and inverse, to changes in their environment. Environmental gradients are used in this process to express how an exposure is increasing in the studied environment. The primary goal in TITAN is to determine if there are levels of exposure along the gradient that influence a statistically significant positive or inverse response and are associated with the presence or absence of one or more specific species. The relationship of each species is assessed via an indicator value that ranges from 0 to 100, with 100 representing a perfect indication of species-specific association with the gradient. The TITAN analysis allows for the consideration of species that have low occurrence frequencies to identify those that possess high sensitivity to the environmental gradient. For example, Khamis et al. used the TITAN methodology to determine how reductions in glacier melting influence the presence and absence of certain aquatic species in rivers and lakes [36–38].

For this study, we defined communities as individual respondents and species as the specific symptoms reported to identify the degree to which each symptom represented a statistically significant indicator of UOGD exposure (CWD, IDW, and AEC). To remove symptoms with frequencies too low to detect a pattern, we only included symptoms reported five or more times into the TITAN analysis (n = 50) [39]. Indicator values were considered statistically significant at an α of 0.05, and resulting symptoms were organized by those having a frequency greater than 10 and a z-score greater than or equal to 1. To our knowledge, this is the first use of TITAN methodology in public health research (S1 Appendix).

## Results

### Symptom reporting characteristics

In this convenience sample of 104 adults who presented health concerns about UOGD, 59% were female with a median age of 57. In this predominantly rural area, only a third reported using municipal water for household use with the majority relying on private wells, cisterns, or springs. Smoking status was available for 78 of the 104; of those, 40% reported either current or former smoking. The number of individual symptoms reported by individuals ranged from 0 symptoms to 36, with mean of 7 symptoms and a standard deviation of ± 7.7 symptoms per person. Table 1 shows the most frequently reported symptoms.

### Generalized linear models: Symptom total

Initial GLMs to test the three spatial scales against symptom total showed that models using 5 km as the radius had the lowest AIC value and were therefore selected in our study (1 km: AIC = 1095.26, 2 km: AIC = 1039.73, 5 km: AIC = 1027.65). Between the three exposure measures, Pearson correlation coefficients ranged from 0.03 to 0.60; thus, all three were tested independently against total reported symptoms. Final GLMs for each exposure measure included sex and smoker status as statistically significant individual predictors, while age was not found to be statistically significant. Sex and smoker status were modeled as categorical

**Table 1. Ten most frequently reported symptoms by number and percent of respondents (n = 104).**

| Symptom | n | n (%) |
|---|---|---|
| Sore Throat | 34 | 33 |
| Headache | 34 | 33 |
| Difficulty Speaking | 34 | 33 |
| Cough | 32 | 31 |
| Itchy or Burning Eyes | 30 | 29 |
| Stress | 30 | 29 |
| Shortness of Breath/Difficulty Breathing | 26 | 25 |
| Anxiety/Worry | 26 | 25 |
| Fatigue | 21 | 20 |
| Sinus Infection | 20 | 19 |

variables, while age was treated as continuous. Water source was excluded during the model selection process and was not included in the final models.

When controlling for age, sex, and smoker status the exposure measures produced the following results: CWD, IDW, and AEC predicted total reported symptoms ($p < 0.001$, $p < 0.001$, $p < 0.05$ respectively). Based on comparisons of AIC values, CWD (AIC = 780.91) appeared to be more closely related to adverse health symptom reporting compared to IDW (AIC = 803.13) and AEC (AIC = 831.95; Table 2; Fig 2).

**Table 2. GLM model results for each exposure variable against total reported symptoms.**

| Model | Variable | Estimate | Std. Error | Z statistic | P value |
|---|---|---|---|---|---|
| CWD | | | | | |
| | Intercept | 1.339 | 0.257 | 5.220 | <0.001 |
| | Ever Smoked | 0.520 | 0.088 | 5.921 | <0.001 |
| | Sex | 0.486 | 0.094 | 5.156 | <0.001 |
| | CWD | 0.840 | 0.102 | 8.267 | <0.001 |
| | Age | -0.002 | 0.004 | -0.605 | 0.545 |
| | Residual degrees of freedom | 73 | | | |
| | AIC | 780.91 | | | |
| IDW Score | | | | | |
| | Intercept | 1.407 | 0.253 | 5.563 | <0.001 |
| | Ever Smoked | 0.492 | 0.088 | 5.615 | <0.001 |
| | Sex | 0.487 | 0.094 | 5.184 | <0.001 |
| | IDW Score | 0.015 | 0.002 | 6.245 | <0.001 |
| | Age | -0.002 | 0.004 | -0.461 | 0.645 |
| | Residual degrees of freedom | 73 | | | |
| | AIC | 803.13 | | | |
| AEC | | | | | |
| | Intercept | 1.508 | 0.250 | 6.029 | <0.001 |
| | Ever Smoked | 0.544 | 0.087 | 6.252 | <0.001 |
| | Sex | 0.550 | 0.094 | 5.855 | <0.001 |
| | AEC | $5.74 \times 10^{-6}$ | $2.35 \times 10^{-6}$ | 2.444 | <0.05 |
| | Age | -0.003 | 0.004 | -0.758 | 0.449 |
| | Residual degrees of freedom | 73 | | | |
| | AIC | 831.95 | | | |

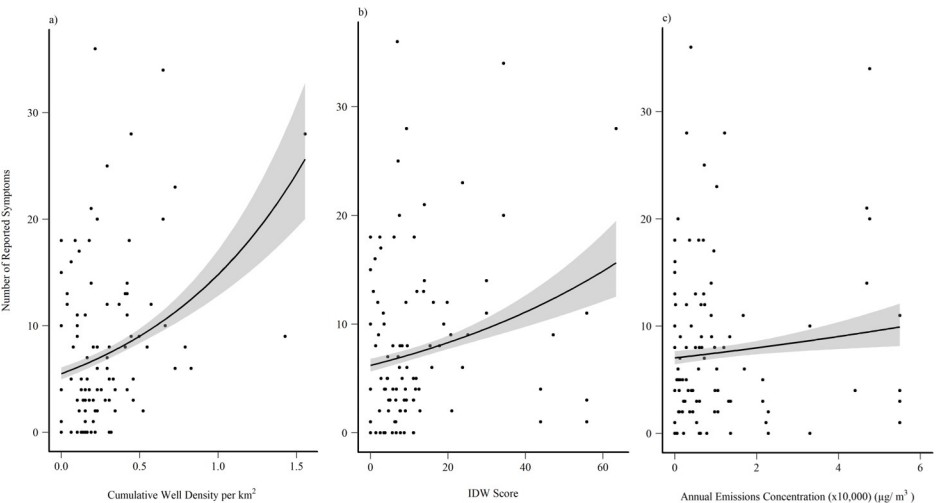

**Fig 2. Exposure model plots.** Poisson distributed generalized linear model for total symptoms and a) CWD, b) IDW score, and c) AEC as the exposure measure. A 95% confidence interval was applied around the regression line.

## TITAN analysis

The TITAN analysis identified multiple statistically significant symptoms along gradients of CWD, IDW, and AEC ($\alpha < = 0.05$). The higher the indicator value, the more likely the symptom is to be seen with an increase in exposure. Twenty-wo symptoms were associated with the gradient of CWD (Fig 3) with itchy or burning eyes as the strongest, positive indicator value along the gradient (indicator value = 59.31), followed by stress (indicator value = 47.17) and dry skin (indicator value = 44.44). Headache, difficulty sleeping, sore throat, stress, and itchy or burning eyes were the five most frequent symptoms in this gradient. Of the twenty-two statistically significant symptoms, approximately, 27% were categorized as EENT symptoms, followed by nerve and muscle symptoms at 27% as well. Four symptoms were inversely associated with the gradient. Although this is counterintuitive, given that 50 symptoms were

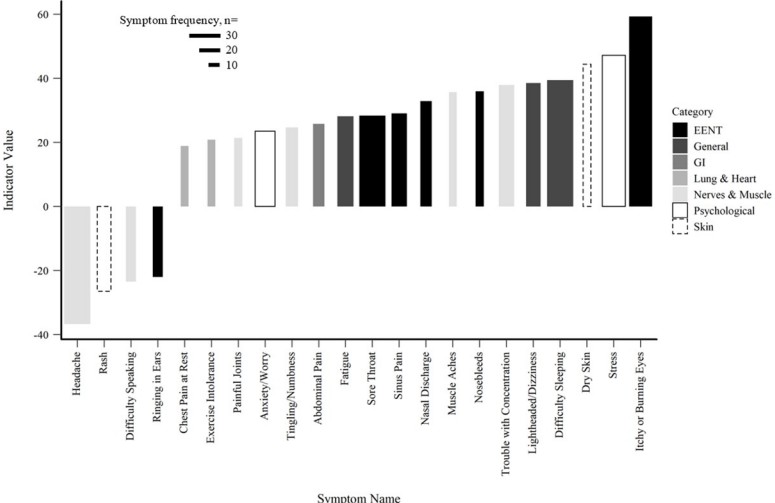

**Fig 3. CWD TITAN results.** Individual symptoms by indicator value along the gradient of CWD. Indicator values range 0–100, with 100 being a perfect association with the gradient. Bar width represents symptom frequency.

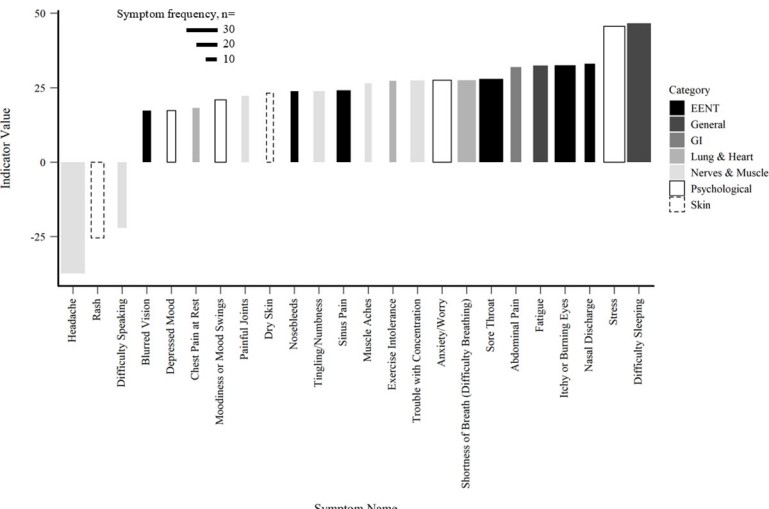

**Fig 4. IDW TITAN results.** Individual symptoms by indicator value along the gradient of IDW. Indicator values range 0–100, with 100 being a perfect association with the gradient. Bar width represents symptom frequency.

assessed along each gradient, one would expect a small number of symptoms be statistically significantly associated with gradients as type-I errors.

Twenty-four symptoms were statistically significantly associated with the gradient of IDW (Fig 4), with difficulty sleeping as the strongest, positive indicator (indicator value = 46.6), followed by stress (indicator value = 45.58), and headache (indicator value = 37.7), though this particular symptom was inversely associated with the gradient. In addition to headache, difficulty speaking, and rash were also inversely associated with the gradient. The top five most frequent symptoms were the same as those in the gradient of CWD. Of the twenty-four statistically significant symptoms, approximately 25% were EENT; 25% were nerves and muscle symptoms; 17% were psychological symptoms.

Seventeen symptoms were statistically significantly associated with the gradient of AEC (Fig 5). Difficulty sleeping represented the strongest, positive indicator value (indicator

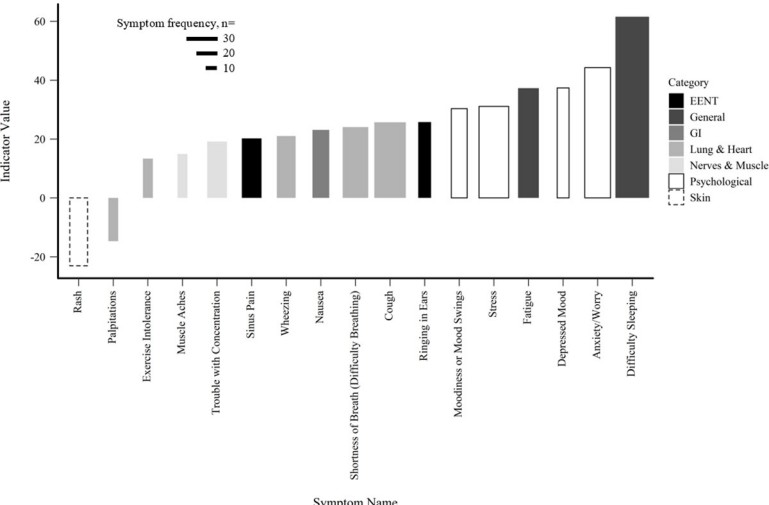

**Fig 5. AEC TITAN results.** Individual symptoms by indicator value along gradient of AEC. Indicator values range 0–100, with 100 being a perfect association with the gradient. Bar width represents symptom frequency.

value = 61.58), followed by anxiety/worry (indicator value = 44.29), and depressed mood (indicator value = 37.36) which were both positively associated. Two symptoms were significantly inversely associated with the gradient of AEC. The top five most frequent symptoms of this gradient were: difficulty sleeping, anxiety/worry, cough, stress, and shortness of breath (difficulty breathing). Of the seventeen significant symptoms, roughly 29% were lung and heart symptoms; 29% were psychological.

## Discussion

Despite a high degree of inherent complexity in associations between health and UOGD, a growing body of evidence, including our findings, suggests that the impacts of UOGD are heterogeneous and consistently detectable even at distances considered safe by some regulations. Determining the best method for quantifying UOGD intensity from a health standpoint is still unknown; however, we detected links between each exposure measure and total symptoms reported, including effects detected at a farther range (5 km) than reported in other studies [15,19]. Variation in UOGD operations can include the size, operation duration, and heterogeneity in chemicals used which adds complexity when attempting to relate operations to health symptoms. Discerning other influences on health that are not UOGD related or interact with UOGD in ways that have not yet been studied is an additional challenge. Other environmental stressors compounded with UOGD, or the inclusion of other UOGD infrastructure like pipelines and compressor stations, further such complexity. The use of amended IDW metrics, such as employed in Koehler et al. [40], attempts to expand IDW by including well development phases to better define exposure. Regardless, the consensus of studies reporting on health impacts around UOGD infrastructure suggests consistency between variables. The aggregate of these analyses suggests that regardless of how exposure to UOGD intensity is quantified, the impacts may occur at broad spatial scales and using distance to just the nearest UOGD facility may underrepresent risks to health.

The method of estimating UOGD intensity appears to affect the strength of associations between exposure and health outcomes in our study, but overall, a positive relationship was found between CWD, IDW, and AEC and total reported health symptoms within a 5-km radius of respondent homes. Brown et al. [23] did not find an association with the median AEC. This apparent inconsistency may be explained by their use of the median AEC, rather than the 90[th] percentile AEC used in this study.

Our model accounts for variation in the results that may be linked to our demographic variables. By doing so, our model terms related to exposure can account for the weight of UOGD after the variability of our demographic variables has been factored out. Relative to AEC and IDW measures, our findings indicate that CWD in proximity to residences, which constitutes a more simplistic measure, was more closely linked to total symptom reporting (Fig 2A). Exposure measures like CWD and IDW are considered proximity metrics and do not define an exact exposure pathway from source to residence; however, we hypothesize that adverse health symptoms could occur through inhalation of chemicals in UOGD emissions and that an increase in the density of wells would, together, create an exposure route. Given that both proximity and a better-defined exposure measure of AEC were significant, future studies should explore links between these measures on their own.

Our challenge to predict adverse health symptoms may reflect the general challenge of condensing well operations into a single, simple metric due to variation in each operation. Studies often apply only one metric for exposure, which could potentially overlook effects that may be seen if the measure were more precise and if more detailed UOGD data were readily available. Regardless of our findings, additional inquiries that compare health outcomes associated with

exposure magnitude coupled with real-time live air monitoring are needed to determine which measure best quantifies exposure.

Our results also caution against limiting investigations of UOGD impacts on health within symptom categories due to the mixed suite of effects reported by respondents. For example, our model assessing the relationship between total symptoms and IDW, and total symptoms with AEC, suggested relatively limited predictability (Fig 2B & 2C). However, the respective TITAN analyses included nearly as many significant symptom associations compared to the CWD model (24 and 17 statistically significant indicators, respectively). Other studies have limited analyses to symptom categories, which may lead to underreporting of impacts to health across the literature, as individual symptoms have been classified under different categories [13,15,41]. A closer look at category composition in other studies revealed that itchy or burning eyes, sinus pain, fatigue, stress, and anxiety/worry are specific symptoms reported by individuals, consistent with our findings in the TITANs [14,15,42,43]. Psychological symptoms, such as stress and anxiety/worry, were included in the top five symptoms either together or separately in each of our models, with the highest percentage of psychological symptoms found in the gradient of AEC. Studies have found that increased air pollution can be linked to psychological distress, while others have found that increased stress, depression, and anxiety can be experienced by people living in communities with UOGD [14,15,42–44]. Furthermore, Albrecht [45] notes that environmental change can cause human distress, which is supported by Lai [46] who found that negative perceptions of UOGD were associated with negative psychological states. The individual symptom counts increased along exposure gradients (Figs 3–5), suggesting subtler effects when compared to aggregate symptom total (Fig 2).

Our results also caution against emphasizing a single symptom to represent detrimental health in association with UOGD. Given the suite of various chemicals applied in UOGD operations and statistically significant interactions between UOGD exposures and demographic variables as highlighted by our GLM models, substantial weight of evidence is needed to conclude that a single symptom is likely to increase with UOGD intensity. The TITAN analyses identified four, three, and two symptoms that were statistically inversely related to the gradients of CWD, IDW, and AEC. Regardless of these anomalies, 18 out of 22, 21 out of 24, and 15 out of 17 statistically significant indictor symptoms were positively associated with the gradients of CWD, IDW, and AEC which contributes further evidence that UOGD impacts health in a heterogeneous manner.

## Limitations & recommendations

As with any work attempting to relate the severity of health impacts to an environmental stressor, our study findings must be considered in the context of the study limitations. Our convenience sample consisted of individuals who presented to EHP because they had concerns about health effects associated with exposure to UOGD, limiting generalizability. Additionally, the health records lacked detailed information about symptoms onset, duration, and severity, or the nature of the symptom (i.e., episodic or chronic). Our lack of detailed information in our symptom data is a limitation of this study. The health records are also subject to recall bias, with the potential for over-reporting of symptoms particularly since respondents presented due to concern about health impacts of UOGD. One mitigating factor is that at the time of reporting their symptoms the respondents did not know their records would be reviewed for this study, nor did they know the exposure measures that would be used. Future studies should collect detailed symptom data and exposure measures in real-time to address these issues.

A further limitation of our study concerns available exposure data. Not all sources of emissions are included in data released by regulatory agencies, and activities such as flaring, off-gassing from contaminated water, and truck traffic may contribute to total emission rates, but are not currently reported [47–49]. In addition, we were limited by available emissions data, which is reported on an annual basis. Some studies suggest that of the development and production stages, the hydraulic fracturing phase of development and the flowback phase of production account for the highest levels of emissions [3,40,50] and future work should include developing exposure measures that capture and isolate these stages.

The air-and-exposure screening model may have also underestimated actual emission concentrations because the model assumes emissions are constant over a year for all sources and does not factor in varying levels of emissions associated with well development phase. Furthermore, our model treats the trajectory of each well's emissions plume equally when summed into one AEC value. Future work should factor wind direction into the model to estimate and correct for the influence wind direction plays on plume movement and concentration to improve upon the AEC value. Additionally, the box model does not correct for influences of topography [25], so we could not compare emission concentrations of various elevations. Regarding weather data, one limitation was that weather data was only taken from one airport for our sample.

## Conclusion

This study was unique in its attempt to use an analytical tool taken from ecological research to determine specific symptom sensitivity to changes in CWD, IDW, and AEC from UOGD. The consistency in relationships between UOGD operations, regardless of how UOGD is quantified, and adverse health outcomes across the literature suggests that increases in symptoms could be related to higher exposure to emissions or chemicals used on the well pad [3,5,11,50]. The impact of fracking on health requires ongoing research because of continued industry growth, the relatively young age of the field, and the potential for chronic or latent illness, like cancer or developmental health impacts, to result from long-term exposure [1,51]. Our results do not confirm direct causal links between UOGD exposure and reported symptoms, but they do suggest that living in proximity to wells may be associated with health symptoms. Our findings suggest that an estimation of exposure that relies only on proximity may be simplistic, particularly in communities with increasing density of wells at 5-km scales, and that a deeper understanding of emissions composition and potency at the residence level is warranted. Future research should examine the question of how the aggregation of exposure affects health.

## Supporting information

**S1 Appendix. TITAN example code and explanation.** Lines 7–13 prepare a sample dataset of twenty potential symptoms and fifty individual respondents to mimic a subset of the data used in this study. For each respondent, 1s and 0s were used randomly for each symptom. A 1 means they did have that symptom, 0 means they did not. Now we have a dataset of fifty respondents and what symptoms they did or did not have. Line 16 creates a randomized list of exposure, one for each of the fifty respondents. In our study, each respondent had a measure of cumulative well density (CWD), an inverse distance weighting (IDW) score, and a measure of estimated annual emissions concentration (AEC). Line 16 creates an exposure variable that ranges from 0 to 50 (no units), with 0 being no exposure and 50 being representative of high exposure, though in our sample there was no limit to how high an exposure measure could go. Line 19 uses titan() to run the TITAN analysis, taking the reported symptoms and exposure values to determine if certain symptoms occur more or less at different levels of exposure. For

example, when the exposure measure reaches 12, the model is looking for any symptoms that stand out as occurring more frequently at that exposure level. Indicator values (range 0–100) are used to score each symptom's relationship to that exposure level, or gradient. A high indicator value shows a strong relationship with the gradient at a certain level. Then, the model determines if that relationship is positive or inverse. In ecological studies, one might study how changes in dissolved oxygen (DO) in a pond ecosystem cause certain species to die off or thrive as levels of DO change. When we begin to see a certain species appear in the pond, we can hypothesize that there may also be a change in DO as well since that species is an indicator of a certain threshold, or level of DO. Lines 22–29 takes information from the TITAN analysis and creates a table. For this table, the rows each represent the different symptoms, while columns are information pertaining to Indicator Value, the frequency of the symptom, p-values, whether the symptom is positively or inversely associated with the gradient, and the z-score. Using these parameters, we begin to filter out symptoms that were infrequent (line 25) and can also filter out insignificant symptoms or symptoms with low z-scores (lines 40–41). The latter two were done in our study but did not make sense for this sample data. Lines 34–36 construct the final plot we used to visualize the results of the TITAN analysis. In the plot, there are ten symptoms positively associated with the gradient with indicator values ranging from 32 to 71. The same goes for the inversely associated symptoms. For the plots in our study, we added additional characteristics like colors to group symptoms into categories and using the width of each bar to represent the frequency of symptoms being reported.
(R)

## Acknowledgments

Dr. Melissa Bednarek, PT, DPT, PhD, CCS (proof reading) and Luke Curtis (proof reading).

## Author Contributions

**Conceptualization:** Hannah N. Blinn, Ryan M. Utz.

**Data curation:** Lydia H. Greiner, David R. Brown.

**Formal analysis:** Hannah N. Blinn.

**Investigation:** Hannah N. Blinn.

**Methodology:** Hannah N. Blinn, Ryan M. Utz, Lydia H. Greiner, David R. Brown.

**Project administration:** Hannah N. Blinn.

**Software:** Hannah N. Blinn, Ryan M. Utz.

**Supervision:** Ryan M. Utz.

**Validation:** Hannah N. Blinn.

**Visualization:** Hannah N. Blinn, Ryan M. Utz.

**Writing – original draft:** Hannah N. Blinn, Ryan M. Utz.

**Writing – review & editing:** Ryan M. Utz, Lydia H. Greiner, David R. Brown.

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
