## [Decision Letter · Decision Letter 0]

21 Jan 2020

PONE-D-19-34629

Exposure assessment of adults living near unconventional oil and natural gas development and reported health symptoms in southwest Pennsylvania, USA

PLOS ONE

Dear Ms. Blinn,

Thank you for submitting your manuscript to PLOS ONE. After careful consideration, we feel that it has merit but does not fully meet PLOS ONE’s publication criteria as it currently stands. Therefore, we invite you to submit a revised version of the manuscript that addresses the points raised during the review process.

While all of the reviewers appreciate the ideas presented by the authors, more than one of them also raised some major concerns. I'd like to give the authors a chance to address the reviewers' comments.

We would appreciate receiving your revised manuscript by Mar 06 2020 11:59PM. To enhance the reproducibility of your results, we recommend that if applicable you deposit your laboratory protocols in protocols.io, where a protocol can be assigned its own identifier (DOI) such that it can be cited independently in the future. For instructions see: http://journals.plos.org/plosone/s/submission-guidelines#loc-laboratory-protocols

We look forward to receiving your revised manuscript.

Kind regards,

Min Huang

Academic Editor

PLOS ONE

Journal Requirements:

Reviewers' comments:

Reviewer's Responses to Questions

**Comments to the Author**

1. Is the manuscript technically sound, and do the data support the conclusions?

Reviewer #1: Yes

Reviewer #2: No

Reviewer #3: Yes

2. Has the statistical analysis been performed appropriately and rigorously? 

Reviewer #1: Yes

Reviewer #2: No

Reviewer #3: Yes

3. Have the authors made all data underlying the findings in their manuscript fully available?

Reviewer #1: Yes

Reviewer #2: Yes

Reviewer #3: Yes

4. Is the manuscript presented in an intelligible fashion and written in standard English?

Reviewer #1: Yes

Reviewer #2: Yes

Reviewer #3: Yes

5. Review Comments to the Author

Reviewer #1: this is a very interesting and well written paper using a technique that is somewhat new for addressing the subject. given that much has been written on the topic, i would suggest stressing your use of statistical modeling to try to better understand the relationships. While you use a convenience sample, the focus should be on testing the model. You acknowledge the limitations of your study, which is refreshing and helpful to the reader.

Reviewer #2: This manuscript looks at the relationship between three UOGD exposure metrics and symptoms among a population of 135 individuals in SW Pennsylvania who approached the Environmental Health Project given their concerns about UOGD. The study adds new techniques, including TITAN borrowed from ecology. They also estimate exposure to UOGD emissions at participant residence, which has been done rarely in the UOGD literature. Poisson models are used to assess associations between the exposure metrics and symptom counts. Models are inappropriately adjusted given the very small sample size and therefore extrapolate beyond the support of the data. Further, the study far oversteps its results in the discussion section. I provide some detailed comments below:

Major

1. The Hess et al. 2019 study had major flaws and I urge you not to cite it. If you want to continue citing it, please describe many of its limitations as discussed by Buonocore et al in their letter to the editor published this month. Koehler et al. also discuss exposure metrics in ES&T in 2018.

2. The number of excluded surveys is very high (~48%). It would be helpful to break this down by reason for excluding the survey, e.g., did most excluded surveys result because the participant did not fully fill it out or because they lived out of state? If most surveys excluded because of incompletion, could you consider doing a sensitivity analysis with this excluded group?

3. I am confused about the first metric. “A cumulative well density was calculated per respondent for the year their survey was completed by taking the total number of wells divided by 5 km.” Wouldn’t you either count the total number of wells within 5km and use that as the exposure or divide the total number by pi*5km*5km (true density of wells). I can’t think of a time you would divide by 5km.

4. Please explain why only a single station was used to derive wind speed and direction. This seems to be a major limitation of the model, given that the study area is over 100km wide. Wind could be dramatically different across this area. If you have reason to believe otherwise, please include. Also, why were emissions of 4 pollutants summed when they can have quite different health effects?

5. What is a model selection and averaging tool? Which demographic variables were considered? In addition, you have a relatively small sample size (N = 135) and it appears you run double stratified models (sex and smoking). This means some models have <33 observations. Given a rule of thumb to have 15 observations for each independent variable, you could really only include the exposure and a single confounding variable in these models. Otherwise, you are extrapolating far beyond your data. Therefore, I am not confident in the current modeling strategy. Further, did symptoms really follow a Poisson distribution or was 0-inflation required?

6. I commend the authors for trying something new but the TITAN analysis is fairly confusing. Consider adding more information to the appendix how why this strategy was selected and how it was implemented, perhaps with a toy example.

7. Lines 327-330: this interpretation of the findings is extremely strong. You have a highly selected sample (people that were worried about UOGD) and a tiny sample size (<200 people). Claiming you have identified which metric to use is over-interpreting your results. Same issue lines 357-360. Tustin et al. had a much larger sample of people who did not enter the study based on UOGD concerns. Therefore, you are asking/answering vastly different questions.

Minor

• Line 69: should this read ambient air pollution?

• Figure 1 add year of active wells to the figure caption

• Please upload higher res figures, these are very difficult to read

• Figure 3: why are bars different widths? If this indicates something, please make clearer what

• Lines 134-136, you say you don’t have data on wells outside PA but then contradict yourself by saying four residences have wells outside PA. If you know about these wells, why not include?

• Line 163: and to be consistent with your other exposure metrics using 5km?

• “An alpha level of <= 0.5 was used as a threshold for significance in both tests” do you mean 0.05?

• First line of results: it wasn’t just a convenience sample but a sample of people reporting issues with UOGD, right?

• What does median age of +/-1 mean?

• Smoking status was never mentioned in methods, please add.

• Results: what was total number of symptoms queried?

• Table 1: how many people included in this table?

• Line 238: which correlation coefficient used?

• How was age modeled? Continuous? What other demographics were considered?

• Lines 316-317 see Koehler et al. 2018 re: compressors

Reviewer #3: Summary of Article

This study uses voluntary health surveys (taken across six years) and data on oil-and-gas (O&G) well locations and annual emissions to identify correlations between reported...[SEE ATTACHMENT FOR FULL COMMENTS]

6. PLOS authors have the option to publish the peer review history of their article (what does this mean?). If published, this will include your full peer review and any attached files.

Reviewer #1: No

Reviewer #2: No

Reviewer #3: Yes: Chris Holder

---

## [Author Response · Author response to Decision Letter 0]

13 Mar 2020

Easier if view in Response Letter submitted as a document. 

Comments Response and Action Taken to Comments Line numbers in revised manuscript where action was taken

Reviewer 1

Overall Comments

This is a very interesting and well written paper using a technique that is somewhat new for addressing the subject. given that much has been written on the topic, I would suggest stressing your use of statistical modeling to try to better understand the relationships. While you use a convenience sample, the focus should be on testing the model. You acknowledge the limitations of your study, which is refreshing and helpful to the reader. This has been addressed. 57-78

Reviewer 2

Overall Comments

This manuscript looks at the relationship between three UOGD exposure metrics and symptoms among a population of 135 individuals in SW Pennsylvania who approached the Environmental Health Project given their concerns about UOGD. The study adds new techniques, including TITAN borrowed from ecology. They also estimate exposure to UOGD emissions at participant residence, which has been done rarely in the UOGD literature. Poisson models are used to assess associations between the exposure metrics and symptom counts. Models are inappropriately adjusted given the very small sample size and therefore extrapolate beyond the support of the data. Further, the study far oversteps its results in the discussion section The model selection tool used for the glm analysis adjusts for the small sample size using a corrected Akaike information criterion (AIC) value, also known as an AICc value. We decided to simplify the glm models for this analysis and remove all interaction terms to only look at main effect variables. Doing so, we can discuss how each metric was related to symptom total after accounting for the other covariates. We have modified the discussion to highlight the issue of small sample size and how that may have influenced our results. 196-206

327-3329

Major Concerns

1. The Hess et al. 2019 study had major flaws and I urge you not to cite it. If you want to continue citing it, please describe many of its limitations as discussed by Buonocore et al in their letter to the editor published this month. Koehler et al. also discuss exposure metrics in ES&T in 2018. This has been addressed and the study removed. NA

2. The number of excluded surveys is very high (~48%). It would be helpful to break this down by reason for excluding the survey, e.g., did most excluded surveys result because the participant did not fully fill it out or because they lived out of state? If most surveys excluded because of incompletion, could you consider doing a sensitivity analysis with this excluded group? Only 23% of survey participants were excluded from the starting sample of 135, as our n for analysis was 104 adults. More detail on exclusion numbers was included. 91-101

3. I am confused about the first metric. “A cumulative well density was calculated per respondent for the year their survey was completed by taking the total number of wells divided by 5 km.” Wouldn’t you either count the total number of wells within 5km and use that as the exposure or divide the total number by pi*5km*5km (true density of wells). I can’t think of a time you would divide by 5km. This has been addressed and updated in the calculations and text. 118-120

4. Please explain why only a single station was used to derive wind speed and direction. This seems to be a major limitation of the model, given that the study area is over 100km wide. Wind could be dramatically different across this area. If you have reason to believe otherwise, please include. Also, why were emissions of 4 pollutants summed when they can have quite different health effects? This has been addressed. We did not have day to day weather data for each residence in our study. Using annually weather data from a local weather station at the Allegheny County Airport was deemed appropriate by the research team since annual averages are more generalizable across airports in the region, compared to day to day values that would certainly fluctuate. We have included this into the limitations section.

You are correct, each pollutant can have different health effects. For this exploratory study, we chose to combine them. Future work will examine individual pollutants. 159-162

5. What is a model selection and averaging tool? Which demographic variables were considered? In addition, you have a relatively small sample size (N = 135) and it appears you run double stratified models (sex and smoking). This means some models have <33 observations. Given a rule of thumb to have 15 observations for each independent variable, you could really only include the exposure and a single confounding variable in these models. Otherwise, you are extrapolating far beyond your data. Therefore, I am not confident in the current modeling strategy. Further, did symptoms really follow a Poisson distribution or was 0-inflation required? The model selection tool uses AICc, a corrected AIC value for small sample sizes. The tool is used to run all the potential statistical models and determine the best fitting models per the AIC values. This does not mean random subsets of data were generated. We simplified the models by removing the interaction terms as possibilities in the final model. 0-inflation was not required for our data as only 15% of the sample reported no symptoms. 196-209

6. I commend the authors for trying something new but the TITAN analysis is fairly confusing. Consider adding more information to the appendix how why this strategy was selected and how it was implemented, perhaps with a toy example We have added supplemental documentation by the way of sample code and a real-world example of TITAN being used in ecology. 223-225

S2 Appendix & submitted R code

7. Lines 327-330: this interpretation of the findings is extremely strong. You have a highly selected sample (people that were worried about UOGD) and a tiny sample size (<200 people). Claiming you have identified which metric to use is over-interpreting your results. Same issue lines 357-360. Tustin et al. had a much larger sample of people who did not enter the study based on UOGD concerns. Therefore, you are asking/answering vastly different questions. This has been addressed. We softened the language and discussed results relative to the characteristics of our sample being small and made up of concerned individuals. We clarified that Tustin et al. did indeed have a larger sample size and therefore isn’t directly comparable. 324-338

363-365

Minor Concerns

Line 69: should this read ambient air pollution? This has been addressed. Throughout the text we have standardized this and similar phrases to “annual emissions concentration” or AEC NA

Figure 1 add year of active wells to the figure caption This has been addressed. 104

Please upload higher res figures, these are very difficult to read Figures were submitted separately to the Journal. When they were included in the PDF submission, the quality was reduced. The Journal will have the highest resolution photos we can provide. NA

Figure 3: why are bars different widths? If this indicates something, please make clearer what This has been addressed. Figure 3 caption explains that the width of the bar is related to the frequency of that symptom being recorded. See the legend on the top right of the image. 278

Lines 134-136, you say you don’t have data on wells outside PA but then contradict yourself by saying four residences have wells outside PA. If you know about these wells, why not include? Fractracker.org provides a mapping tool of wells outside of PA but does not give Latitude/Longitude of these wells or provide a downloadable dataset for us to map on our own. We could type in respondent’s addresses and see that there may be wells within their buffer but could not apply this data to what we did in ArcGIS. 

We only used PA gas wells so removed reference to wells outside of the state since we are not able to quantify their emissions, and yes, this would underestimate their exposure. Furthermore, only 7 out of 104 subjects we assessed were near enough to state borders to result in a reduced areal extent, and these possessed sample areas >50% within PA. 130-131

Line 163: and to be consistent with your other exposure metrics using 5km? This has been addressed. 117

“An alpha level of <= 0.5 was used as a threshold for significance in both tests” do you mean 0.05? This has been addressed. 193

First line of results: it wasn’t just a convenience sample but a sample of people reporting issues with UOGD, right? This has been addressed and more language about the sample characteristics has been added. 237

What does median age of +/-1 mean? This has been addressed. 238

Smoking status was never mentioned in methods, please add. This has been addressed. 199

Results: what was total number of symptoms queried? 779 NA

Table 1: how many people included in this table? This has been addressed. 243

Line 238: which correlation coefficient used? Pearson – this has been addressed in the text. 246

How was age modeled? Continuous? What other demographics were considered? Continuous – this has been addressed in the text. 249-250

Lines 316-317 see Koehler et al. 2018 re: compressors This has been addressed. 318

Reviewer 3

Overall Comments

It is particularly interesting that the correlation with total reported symptoms is stronger with CWD than with IDW, and the authors make a good case for why that is—that being, the authors did not have data on the particulars of each well’s activities (beyond location and annual emissions), and assigning higher weights to wells closer to a residence compounds that uncertainty. In my opinion, that points out a notable limitation in the methodology and conclusions of this study, which I believe the authors should be more straightforward in acknowledging. That is, the health concerns reported by residents are correlated only with annual data on well locations and emissions. It is not known if the health issues were transient or longer lasting (which the authors acknowledge), and it is not known exactly what was going on at the well pads within 5 km of their house. We know that wells under development can have highly variable emissions, perhaps by orders of magnitude, and some wells may only be under development for weeks before going into production mode, during which emissions are generally much smaller. The body of literature suggests that higher air concentrations resulting

from O&G activities are much more likely to occur during development, and that reports from local residents of health issues and nuisances also tend to peak during development. Therefore, correlating one-time reports of health complaints with annual O&G data is missing an opportunity to more directly investigate possible connections between health complaints and

O&G operations in real time. Can you say if new well development is active and thriving in these counties, mixed with wells in long-term production mode? Are there really no data on which wells were under development vs. in production, with sufficient time resolution to draw closer connections? We have added language in several sections to be clearer about these specific limitations. We acknowledge the annual emissions concentration does not account for day to day exposure at the household, given the different stages of well development have different lengths and emissions amounts. We added additional citations to support this claim. Any inclusion of well pad activity or development would have been an estimate on our part, as we do not have access to the start and end date of the various activities that occur on the well. Our dataset only including the spud date for the well. We do know that in these counties, wells are continuously being drilled and same are being re-fracked. That is not tracked on a public database for us to use.

 60-61

159-162

180-182

333-343

379-387

391-397

And if you are asserting that these health-symptom reports may be linked inhaling chemicals emitted from well pads, then emissions and proximity are key to that exposure route. However, you should be clearer in your assumption that the respondents’ exposures are entirely at their residence (or at least that’s what this intensity metric represents) and that there is full chemical penetration into their home. I also found the “Ambient Air Emissions” methodology section to be rather unclear on a number of fronts, as I discuss in more detail below. This is the section that I advise the most revisions to. We added additional language to add clarity to this assumption and to be clear the exposure metric is calculated at the residence. 180-182

333-338

416-419

I also appreciate that the authors used several characteristics of respondents as part of the GLM application. However, the discussion of the role of those characteristics in correlating symptom reporting with changes in exposure intensity is non-existent. In my opinion, if you are making observations about statistical differences in males/females, smokers/non-smokers, age,

water source, etc., then the discussion on them should be more complete and include

speculations about the meaning of those differences. If you are not willing to speculate, then say why. The adjustments we made to simplifying the glm models are meant to also address this point and we added language to the discussion about how our models do account for the covariates like sex, age, and gender. Since we are controlling for the covariates, our discussion focuses on the relationship between exposure metrics and symptom total only (see revised Table 2). Starting at 245 under “Generalized Linear Models: Symptom Total”. 

327-329

Major Concerns

Study Sites & Health Outcomes 

Line 90: What does “Appendix A” refer to? The reference doesn’t have an Appendix A, and neither do you? Also, the link provided for reference 18 is broken, I think you’re missing a hyphen between “individual” and “heath”. An appendix has been added and supplemental materials provided separately. S1 Appendix

Cumulative Well Density and Inverse Distance Weighting 

Line 119: You say that three radii were drawn initially. That implies something changed later…? This has been addressed. For this study, we only chose to look at wells within a 5-km buffer. The 1-km and 2-km buffers were initially included in the text to explain why we ended up choosing 5 km from a statistical approach (comparison of AIC value between buffer distances and CMD), but we have elected to instead site Elliot et al. (2019) as they also looked at a 5-km buffer range. We removed reference to the smaller buffers as we did not run glm analyses on these distances. 118-120

Line 122: suggest updating this sentence to “Active, unconventional wells for the year of a completed health assessment were plotted within the three radii around the respondent’s home.” Though 5 km was the maximum radius, it’s probably more clear to say within the three radii. This has been addressed. 118-120

Line 123: update this sentence to “A cumulative well density was calculated per respondent for the year of their survey, equal to the total

number of wells divided by the radius (in km).” Again, it’s using three radii with 5 km being the max, right? This has been addressed. 118-120

Line 135 about the four residences with wells outside of PA within their 5-km radius (insert hyphen there!): were they also outside of PA for their 1- and 2-km radii? Also, why not throw these respondents out of your study since you essentially had to remove part of their buffer areas, making their exposure measures underestimates? We only used PA gas wells so removed reference to wells outside of the state since we are not able to quantify their emissions, and yes, this would underestimate their exposure. We added information to describe the percent of these participant’s 5-km buffer that was in West Virginia. 130-131

Ambient Air Emissions 

“Ambient Air Concentrations” is a more appropriate section title, as those are what you are deriving in this section. This has been addressed. Annual Emissions Concentration was the phrase chosen to standardize how we referred to this measure. Thank you for your input on this. 133

Line 139: should year 2012 been removed from your assessment, given that 25% of the year’s emissions data were unavailable? It is our understanding that all 2012 data was reported to the PA DEP. Given the industry uses algorithms to estimate their yearly emissions, we question whether they would adjust the formula to exclude months from their calculations. If confusion is coming from Brown et al.’s statement “Since March 31, 2012, owners and operators of natural gas production and processing operations have been required to report air emissions to the PA DEP…” we want to point out that this statement does not reflect a belief that only data after March 2012 was included. 

Brown DR, Greiner LH, Weinberger BI, Walleigh L, Glaser D. Assessing exposure to unconventional natural gas development: using an air pollution dispersal screening model to predict new-onset respiratory symptoms. Journal of Environmental Science and Health, Part A. 2019 Dec 6;54(14):1357-63. NA

Line 155: for the Pittsburgh meteorological data, can you speak to their representativeness of conditions across your study area? This has been addressed. We did not have day to day weather data for each residence in our study. Using annually weather data from a local weather station at the Allegheny County Airport was deemed appropriate by the research team since annual averages are more generalizable across airports in the region, compared to day to day values that would certainly fluctuate. We have included this into the limitations section. 159-162

392-393

Line 163: what is the significance of modeled concentrations being less than 10 μg/m 3 , and what was that based on? This has been removed. The rational that we were keeping a buffer of 5-km across our metrics serves as a clearer justification. NA

Lines 171–173: an air concentration is not a “rate of emissions exposure”, it’s a concentration. You can then say (if correct) that you assume that it is the concentration at which residents are exposed (i.e., constant exposure to outdoor concentrations at their residence), and refer to it as an exposure concentration. What is meant by “total, or aggregated, emissions”—the one emissions rate you say earlier that you used in your calculations? Repeating that here makes it sound like there’s something more going on here at the end to determine individual exposure. This has been addressed. See section “Annual Emissions Concentration” starting on line 133

There is an inconsistency in terminology that can be easily rectified: you are using emissions of chemicals from well sites (which indeed are ambient air emissions, though “air emissions” is

clear enough; and they are reported as rates, not volumes; a consistent use of emission “rate” rather than “amount” is desired here, too) and meteorological data to estimate concentrations (not “levels” or “emissions concentrations” or “air level value”) of those chemicals in the ambient air at various distances from the well. This has been addressed. Emissions concentration is the phrase we selected to replace the varying ways we previously were describing it. See section “Annual Emissions Concentration” starting on line 133

You kindly offer a brief summary of a box-model methodology more fully described in other papers, but the brief summary as it is currently written is inadequate and confusing. Pasquill used five wind-speed categories, along with cloud cover and time of day, to define six stability classes, not 30. I see that your reference [23] (Brown et al., 2019) has a Table 1 that defines 30 stability classes from A1 to D30, but I don’t recall seeing these from Pasquill’s work (certainly correct me if I’m wrong!) and I don’t see how that’s used in concert with Figure 1 of [23] which is the vertical mixing/stability/distance look-up chart showing just the six stability classes. You say you pulled data on wind direction from NOAA, but the box model does not utilize wind direction. Taking a step back from the details, it would be clearer to say (roughly) that the model utilizes atmospheric stability, wind speed, and an

assumption about the size of a well-pad facility to estimate the size of a box in which the emissions are well mixed, which in turn is a measure of plume dilution, where the chemical concentration in the box is calculated as emission rate divided by box volume. (Hourly wind speed is used as part of the box volume calculation, right? That’s the “meters of air that pass over a site/minute” stated in [23]?) Then you can march through how you identified each of those parameters (Pasquill stability from hourly data on cloud cover and wind speed from NOAA; an assumed 100-m diameter of well pad; Pasquill assumptions on vertical mixing given stability

and horizontal distance; an assumption of constant 300 g/h emissions). This has been addressed. We believe the confusion was coming from the use of stability classes and stability conditions, where they are properly called stability classes and meteorological conditions, based on solar radiation, that define correlated stability classes. In Brown et al. (2019), five wind speeds were used alongside 6 meteorological conditions to create the 30 correlated stability classes that we applied to hourly weather data from the Allegheny County Airport (their Table 1). Pasquill’s original correlated chart shows 5 meteorological conditions, but Brown et al. 2019 added a fourth daytime insolation category. 

Brown DR, Greiner LH, Weinberger BI, Walleigh L, Glaser D. Assessing exposure to unconventional natural gas development: using an air pollution dispersal screening model to predict new-onset respiratory symptoms. Journal of Environmental Science and Health, Part A. 2019 Dec 6;54(14):1357-63. 144-162

The relationship between the reference emission rate used in the modeling (300 g/h) and the actual facility emission rates (variable)

is not entirely clear. I think you’re telling me that you ran the model to get concentrations per unit emissions at five different distances

from a well, based on a high-end metric of hourly concentrations in a year (why 300 g/h and not just 1 g/h?). Then you got a well’s real emissions and multiplied them by the modeled concentration per unit emissions. If that is correct, please consider updating the

final paragraph of this section to be more clear about this. This has been addressed. You are correct, the 300 g/h is arbitrary as any hypothetical emissions number, above zero, would give the concentration value at each distance. The model was run to determine the concentration at each distance from the well. When we determined the distance between the well and a home, we took that concentration value, multiplied it by the emissions rate from the DEP dataset, and then divided by 300 to get what is estimated to be the emissions concentration at that residence µg/m3. 

 163-172

The use of quadrants around a residence is not clear to me. How does this affect concentration in any way? This has been addressed. We factor in what directional quadrant the well is from the home and the appropriate 90th percentile concentration value for that quadrant and the distance from the home is chosen and used to calculate the estimate emissions from a given well. Then we add the concentrations calculated from the North, South, East, and West into one AEC value. The quadrants are a step in the calculation. 173-182

Statistical Analysis 

Lines 184 and 219: was the threshold of significance 0.5 or 0.05? This has been addressed. 193

Final paragraph: what is the purpose of only including symptoms reported 5+ times, and organizing symptoms with frequencies > 10? We used these exclusion parameters to remove symptoms that are too infrequent to do statistical analysis with. The numbers are arbitrary and simply provide a way for us cut out infrequencies. 228-230

Generalized Linear Models: Symptom Total 

I must admit that Table 2 is difficult to follow. Would a strong statistician understand this? Under a given model, should I be looking to the line restating the model name as the variable, to find statistical significance (e.g., p<0.001 for Cumulative well density-Cumulative Well Density and for Aggregated emissions-Aggregated Emissions, but p=0.316 for IDW Score-IDW Score)? What’s the meaning of the “Cumulative Well Density: Age” type rows, and why is there only one of these kinds of rows for the IDW Score model, while three for Aggregated emissions model and five for Cumulative well density model? The removal of the interaction terms (Cumulative Well Density: Age) were removed to simplify these models. Table 2 has been updated to reflect these changes. Removing the interaction terms and adjusting the IDW metric as explained in the cover letter let to the IDW metric to be a significant predictor in the study and this has been updated throughout the text. 

In the table, the name of the model is given in italics. That is identifying with exposure model was being testing against symptom total (while adjusting for our demographic variables). When you read the table, you will see Cumulative Well Density statistically significantly predicted Symptom Total while adjusting for covariates sex, age, and gender. The model was significant at a p<0.001. Citation – 230

Table 2 – 256 

Paragraph starting Line 250: I’m not sure what the second sentence is saying, the part about “occurred between smoker status”. It appears to

me that there was little-to-no relationship between number of symptoms reported by current/former male smokers and magnitude of emissions, and for males not reporting their smoker status the model actually shows

a declining number of reported symptoms and as emissions increased. I think your declarations of these results could use qualifying terms like

“generally” or “on average”: “Increased symptom reporting also generally occurred…”, “Females on average also reported more symptoms than males…”, and so on. Change “with” on lien 254 to “between”. Final sentence: you probably should say that water source isn’t shown in Figure 2, for clarity? However, again going back to my confusion over Table 2, doesn’t it show p=0.049 (which is >0.001) for Cumulative Well Density: Water Source, and thus isn’t statistically significant? This has been addressed by simplifying the models to only show main effects. See Table 2. NA – text was omitted 

Figure 2b’s X axis is better labeled as concentration rather than emissions, as covered earlier in my comments. This has been addressed. NA

TITAN Analysis 

Lines 281–282: range and mean was the same as what? Same on line 295. This has been addressed and the confusing text removed to simplify the presentation of the results. NA

Figures need y-axis labels This has been addressed. NA

Minor Concerns

“Household” refers to the people in the house. I think in most cases you mean “residence” (i.e., the location of the house). This has been addressed. NA – updated throughout text

In many cases, you use “gas well” as shorthand for “oil and gas well” but it implies by omission that they’re not oil wells. Consider just using “well”. This has been addressed. NA – updated throughout text

If you estimate concentrations from emissions, then consider if your model names and

results discussions should refer to a “concentration model” and “concentration intensity” and “concentration gradient” etc. rather than emissions model, intensity, gradient, etc. This has been addressed. NA – updated throughout text

Not defining duration of symptom (short periods vs long periods of symptom persistence) is a concern in terms of understanding if reported health issues are episodic versus chronic. Not correlating time of symptom with UOGD activity also weakens assumptions about correlations between well activities and health issues. This has been addressed.

 380-382

I take some issue with calling the reported symptoms “health effects”, “health impacts”,

and similar phrasing. These phrases imply cause (O&G emissions) and effect (itchy eyes, etc.). Perhaps terms like “negative health symptoms” are more appropriate? This has been addressed. Negative health symptoms used instead. NA – updated throughout text

I also think you should be more careful about referring to CDW and IDW as measurements of exposure. They’re metrics of proximity to wells, and that’s it. CONC is closer to an exposure metric, as you attempt to estimate air concentrations of O&G-emitted chemicals, at residences. I think at the least you should acknowledge this, and perhaps then establish that for convenience you will refer to them as metrics of potential exposure intensity (or something like that). This has been addressed and we clarify that CDW and IDW are proximity in the introduction but will be referred to as exposure metrics. In the discussion we re-discuss this distinction. 60-61

333-336

Abstract 

Lines 26–27: suggest rewording sentence to: “We investigated UOGD density and well emissions and their associations with symptom reporting by residents of southwest Pennsylvania.” This has been addressed. 24-25

Line 28: change “from 2012-2017” to “in 2012–2017” (en dash) This has been addressed. 26

Line 31: insert comma after “intensity” This has been addressed. NA – text modified or removed

Line 34: change “ambient air emissions” to “ambient-air emissions” This has been addressed. NA – text modified or removed

Line 35: a dispersion model does not quantify emissions This has been addressed. 34-35

Line 41: change “comprised of” to “constituted” This has been addressed. 39

Line 42: I think you should change “increased” to “increases”? This has been addressed. 41

Introduction 

Lines 48–49: change “human health risk” to “human-health risk” This has been addressed. 46

Line 61: change “number” to “numbers” This has been addressed. 59

Lines 67–68: insert comma after the [8] citation, and change “inverse distance weighting” to “IDW” This has been addressed. 67

Lines 69–70: I think you should change “well emissions exposure” to “emissions exposure metric”? Also, change “calculate ambient air at the” to “calculate ambient-air concentrations at the”. Change “exposure metric comparison as well, however, their” to “exposure-metric comparison as well, but their” This has been addressed. 68-69

Line 72: I think you should put “[16]” after the Hess citation? This has been addressed and the study was removed. NA – text modified or removed

Lines 74–75: suggest revising as “...and this analysis—comparing three estimates of exposure, including reported emissions—attempts…” This has been addressed. 71-72

Final sentence starting on Line 78: suggest changing to “The aggregate of methodologies applied here—using statistical modeling to analyze the influence of different exposures on symptom reporting, and applying a technique to identify specific symptoms that might be indicative of exposure—is novel in UOGD research and provides insight into new techniques for studying relationships between health and exposure variables.” This has been addressed. 75-78

Study Sites & Health Outcomes 

Line 87: change “Between” to “In” This has been addressed. 83

Line 95: I think you should change “Weinberger et al. the” to “Weinberger et al. [19], the”? This has been addressed. 87,91

Line 97: change “oil and gas industry” to “oil-and-gas industry” This has been addressed. 96

Line 98: suggest changing “complete the assessment form (n=118). The 118 health assessments” to “acomplete the assessment form (17 excluded). The remaining 118 health assessments” This has been addressed. 97-101

Line 99: change “health care providers” to “health-care providers”, and “occupational health physician” to “occupational-health physician” This has been addressed. 92

Line 103: change “one of eight counties” to just “eight counties” This has been addressed. 100

Figure 1 caption: remove comma in “Southwestern, PA”; change “Lawrence county” to “Lawrence County”; insert “County” after “Butler”; change the “[20]” citation to “[22]” This has been addressed. 103-105

Figure 1: suggest making county names more readable (move them on top of the well locations?) This has been addressed. New map image included with submission. NA

Cumulative Well Density and Inverse Distance Weighting 

Citation numbering got messed up in a few spots. I think [20] in the first paragraph should be [21], and in the second paragraph [22] should also

be [21] while [20] should be [22]. Please check. Also in the second paragraph, I think you should insert “[19]” after the Weinberger citation. This has been addressed. 111,113,115

Line 120: change “1km” to “1 km” (insert space) This has been addressed. NA – text modified or removed

Line 126: the “IDW” abbreviation was already established earlier. This has been addressed. NA – text modified or removed

Line 127: should “qualifying” be “quantifying”? Leading into the next line, change “closer to the respondents’ home” to “closer to a respondent’s

home”. This has been addressed. NA – text modified or removed

Line 128: suggest updating this sentence to “The inverse distance of each well within 1-, 2, and 5-km radii of a residence was calculated, and those

values were summed into one IDW score per respondent, per radius, as shown in the following equation:” This has been addressed. NA – text modified or removed

Lines 132–133: change “respondents’ home, and n is the number of wells within the 5 km buffer” to “respondent’s home, and n is the number of

wells within the radius” This has been addressed. 127

Ambient Air Emissions 

Line 139: why does it matter that emissions data after December 31, 2017 were unavailable? The health survey data stopped after 2017 anyway. This has been addressed. NA – text modified or removed

Line 140: insert hyphen: “emissions-inventory data” This has been addressed. 135

Line 141: define PM 2.5 This has been addressed. 136

Line 142: reference [22] should be [21] This has been addressed. It is back to being [22] simple to do the removal and addition of citations in the reference list, but [22] reflects the citation for the PA DEP data. 137

Lines 144–146: what emission sources weren’t included? You mention this much later, but would be good to mention here, too. This has been addressed. 141-142

Line 154: “that was collected at the Pittsburgh Allegheny County Airport” is more clearly stated as “for the Pittsburgh Allegheny County Airport” This has been addressed. 157

Line 156: your study was through 2017, not 2016. This has been addressed. 162

Line 162: insert hyphen: “16-km radius” This has been addressed. NA – text modified or removed

Statistical Analysis 

Line 181: insert comma after “intensity of each exposure” This has been addressed. 190

Line 185: insert “for” after “were used”? This has been addressed. 195

Line 186: change “a model selection and averaging tool” to “a tool for model selection and averaging” This has been addressed. 196

Line 187: change “best fit” to “best-fit” This has been addressed. 198

Line 189: change “one hundred” to “100” This has been addressed. 200

Line 190: I’m not sure what “to model assessment” means, and why 100

models? 100 is the default number of models that are run prior to the fitting of the best models. Out of 100 models, the model with the lowest AIC value is considered the best combination of variables. 205-206

Line 194: change “model” to “models” This has been addressed. 204

Line 202: insert comma after “emissions” This has been addressed. NA – text modified or removed

Symptom Reporting Characteristics 

Line 227: is the median age better stated as “57 ± 1 standard deviation (SD)”? Note to put a space after ±. If this is the correct expression, then consider also updating lines 230–231 to say “with a mean of 7 ± 7.7 SD”. This has been addressed. NA – text modified or removed

Lines 229–230: is this sentence saying that some respondents reported 0 symptoms, while one or more respondents reported 36 symptoms, with an average of 7 reported per person? Consider rewording to be more clear. This has been addressed. 241-243

Generalized Linear Models: Symptom Total 

First sentence: consider rewording to “Based on the initial GLMs (including demographic variables), models using a 5-km radius for

cumulative well density had the lowest AIC value (relative to the 1- and 2-km radii), and 5 km was therefore used as the defining radius for

cumulative well density as an exposure variable…” This has been addressed. NA – text modified or removed

Line 238: change “from 0.05 to 0.73, thus all three” to “from 0.05 to 0.73; thus, all three” This has been addressed. 247

Figure 2a’s X axis should have a space between 5 and km (“5 km”). This has been addressed. NA

Line 248: insert “did” before “emissions” This has been addressed. NA – text modified or removed

Line 250: revise to “In the cumulative-well-density model…” This has been addressed. NA – text modified or removed

TITAN Analysis 

Lines 264–265: update “along the cumulative well density, IDW, and emissions gradients” to “along gradients of cumulative well density, IDW, and emissions”. Otherwise, proper grammar would suggest that you say “along the cumulative-well-density, IDW, and emissions gradients”, which is fine but some consider to be awkward. Consider this throughout this section (e.g., like 267 “cumulative-well-density gradient” or “gradient of cumulative well density”). This has been addressed. Language corrected starting at 262 and going through the “TITAN Analysis” section.

Figure 3 discussion: Suggest you introduce the figure on line 266: “Fig 3 shows results for cumulative well density, with the 23 significant

symptoms displayed. Itchy or burning eyes…”. Then later on line 270, you can remove the “Of the twenty-three significant symptoms, ” preface and just begin “Roughly 26% of the symptoms were categorized…”. This has been addressed. 269

Lines 272–273: What is the meaning of negative associations? Increasing well density leading to decreasing reports of headaches, difficulty

speaking, ringing in ears, and rash? That’s odd, isn’t it? It might be worth mentioning your “type-I error” hypothesis here? Earlier on line 266, you

when discussing the top three indicator values, you probably should include that they’re positive associations, just to be clear? These negative associations are considered type-I errors in our analysis. It does not necessarily mean that was density increased headache decrease, but rather we expect with so many symptom variables being thrown into the model, some will return as anomalies/errors. 271-274

369-373

Lines 272 and 281: should you use “negatively associated” or “inversely associated”? This has been addressed. NA – text updated in multiple places in section

Lines 284–285: maybe you mean “followed by nerves and muscle symptoms and psychological symptoms, which comprised 21% of

symptoms each”? This has been addressed. 286

Lines 285–286: This sentence might be better as “In addition to headaches, difficulty speaking was also negatively associated with the gradient.”? Similarly, lines 294–295: “In addition to headaches, rash and palpitations were also negatively associated with the gradient.” This has been addressed. 282

Lines 298–299: end of sentence might be better stated as “with psychological symptoms and nerve and muscle symptoms each at 20%.”? This has been addressed. 299

Discussion 

Lines 309, 314, 318, and 323: “human-health standpoint”, “human-health symptoms”, “human-health impacts”, “human-health metrics” (insert hyphen) This has been addressed. NA – text updated in multiple places in section

Line 311: I’m not 100% certain but I think “further” should be “farther”? This has been addressed. 311

Line 314: change “non-UOGD” to “not UOGD” This has been addressed. 314-315

Line 320: remove “that” This has been addressed. NA – text modified or removed

Line 322: change “does” to “do” This has been addressed. NA – text modified or removed

Line 325: “5-km” (insert hyphen) This has been addressed. 326

Line 334: change “was” to “were” This has been addressed. NA – text modified or removed

Lines 335–336: “exposure-magnitude impacts (insert hyphen) This has been addressed. 343-344

Line 346: I think the “which may lead to underreporting of impacts to health across the literature” should be bounded by commas This has been addressed. 351-352

Lines 351–352: consider changing “in each model” to “in each of our models” for clarity This has been addressed. 357

Lines 365–366: consider “related to the gradients of cumulative well density, IDW, and emissions, respectively”, rather than “related to the cumulative well density, IDW, and emissions gradient respectively”. Similarly later on line 370:

“associated with gradients of cumulative well density, IDW, and emissions, respectively, which contributes” This has been addressed. 355, 370

Last sentence: why would these aspects lead to underestimates of emissions? This has been addressed. 398-402

Limitations & Recommendations 

Lines 375–376: “health-reporting data” (insert hyphen) This has been addressed. NA – text modified or removed

Line 378: should the Tustin and Rabinowitz citations have numbered indicators

(and Elliot on line 379, Rabinowitz on line 385)? This has been addressed. NA – text modified or removed

Lines 391–392: consider revising sentence to “In future studies, other health indicators or metrics such as blood pressure, heart rate, and the number of days a symptom persisted could provide a more in-depth analysis and help define the severity of the symptoms experienced.” This has been addressed. NA – text modified or removed

Line 400: change to “The air-and-exposure screening model” (insert hyphens) This has been addressed. 398

Conclusion 

Line 408: remove extra space before “an IDW metric” This has been addressed. NA – text modified or removed

Line 410: “detrimental health” is awkward, do you mean health complications? This has been addressed. Yes, health complications. 409

Line 414: insert “the” before “relatively young age” This has been addressed. 412

Line 416: change “but do raise” to “but they do raise” This has been addressed. 415

Line 418: change “oil and gas industry” to “oil-and-gas industry” This has been addressed. NA – text modified or removed

Lines 418–419: suggest rewording to “Our study suggests that the narrow consideration of exposure risk used in some locations—based only on proximity to an individual well—may warrant revision to assess the contribution of the

growing density of wells in and around communities at 5-km scales.” This has been addressed. 416-420

---

## [Decision Letter · Decision Letter 1]

7 Apr 2020

PONE-D-19-34629R1

Exposure assessment of adults living near unconventional oil and natural gas development and reported health symptoms in southwest Pennsylvania, USA

PLOS ONE

Dear Ms. Blinn,

Thank you for submitting your manuscript to PLOS ONE. After careful consideration, we feel that it has merit but does not fully meet PLOS ONE’s publication criteria as it currently stands. Therefore, we invite you to submit a revised version of the manuscript that addresses the points raised during the review process.

As recognized by the reviewers, the manuscript has been much improved. The reviewers have further questions related to your methodology and discussions, and suggest to improve the presentation (language and table/figure quality). Please take this opportunity to address the reviewers' remaining comments.

We would appreciate receiving your revised manuscript by May 22 2020 11:59PM. To enhance the reproducibility of your results, we recommend that if applicable you deposit your laboratory protocols in protocols.io, where a protocol can be assigned its own identifier (DOI) such that it can be cited independently in the future. For instructions see: http://journals.plos.org/plosone/s/submission-guidelines#loc-laboratory-protocols

We look forward to receiving your revised manuscript.

Kind regards,

Min Huang

Academic Editor

PLOS ONE

Reviewers' comments:

Reviewer's Responses to Questions

**Comments to the Author**

1. If the authors have adequately addressed your comments raised in a previous round of review and you feel that this manuscript is now acceptable for publication, you may indicate that here to bypass the “Comments to the Author” section, enter your conflict of interest statement in the “Confidential to Editor” section, and submit your "Accept" recommendation.

Reviewer #2: (No Response)

Reviewer #3: (No Response)

2. Is the manuscript technically sound, and do the data support the conclusions?

Reviewer #2: Partly

Reviewer #3: Yes

3. Has the statistical analysis been performed appropriately and rigorously? 

Reviewer #2: Yes

Reviewer #3: Yes

4. Have the authors made all data underlying the findings in their manuscript fully available?

Reviewer #2: No

Reviewer #3: Yes

5. Is the manuscript presented in an intelligible fashion and written in standard English?

Reviewer #2: Yes

Reviewer #3: Yes

6. Review Comments to the Author

Reviewer #2: Thank you for your thorough responses to the last round of edits, especially during this difficult time. I have a few remaining comments.

A few questions related to Brown et al. 2019 (the exposure metric paper).

1. Why does the present study omit formaldehyde but it was included in the earlier study?

2. The final sample size ended up being 87 in the 2019 paper, were you able to acquire additional data?

3. What is the distribution of years that the participants took the survey in the present study? No emissions data was available in 2017, so did these subjects receive estimates from 2016?

4. Switched from 2 to 5km buffer between the two studies, what was the rationale?

Other comments

1. Please add some explanation regarding the decision to combine 80+ (in many cases completely unrelated) symptoms into a single symptom count variable for the regression analysis.

2. How does the present study compare to Brown et al. 2019 where only respiratory symptoms were assessed and no association was found with air emissions? Please add some text to the discussion section on this topic.

3. The figures continue to be too low resolution to really read. Please update these to at least 300 dpi.

4. Add n (%) to table 1, please.

5. Discussion, page 21, line 429+: the CWD metric is most strongly associated with total symptom count. To me, this indicates a physical and psychological pathway between UNGD and health.

Some discussion of how perceived environmental change is associated with health might be helpful here. For example, the idea of solastalgia, feeling homesick even at home due to changed environmental conditions around the home (Albrecht). Further, in Lai 2017 “Understanding the psychological impact of unconventional gas developments in affected communities,” the authors find that negative perceptions of unconventional development was associated with negative psychological states.

6. It should be emphasized that this study provides evidence that among people aware and concerned about UNGD the CWD metric performs best, we have no idea if this would be the case with people not concerned or pro-UNGD. There is an interplay of perception, psychology, and health that should be more carefully discussed.

Reviewer #3: See comments in attachment.

Thank you for the opportunity to review your revised manuscript. Your revisions have satisfactorily addressed most of my concerns from my review of your initial submission. As with my first review, I currently have numerous minor concerns related to proofreading. Aside from those, I have a small number of less minor concerns related to your methodology, and one related to your conclusions, that should be addressed within the paper before I can recommend publication.

7. PLOS authors have the option to publish the peer review history of their article (what does this mean?). If published, this will include your full peer review and any attached files.

Reviewer #2: No

Reviewer #3: Yes: Chris Holder

---

## [Author Response · Author response to Decision Letter 1]

18 May 2020

Please see cover letter for best formatting of responses: 

Comments Response

Reviewer 2

1. Why does the present study omit formaldehyde but it was included in the earlier study? Brown et al. (2019) calculated emissions from wells, processing plants, and compressor stations. Brown et al. established the 5 compounds with the highest reported mass and known health effects. This analysis calculated emissions from wells only; this analysis did not include formaldehyde because it was not one of the top 5 compounds emitted from wells. 

See lines 140-147

2. The final sample size ended up being 87 in the 2019 paper, were you able to acquire additional data? The 2019 paper excluded those whose “residence was outside Pennsylvania and whose residence was outside of the county of interest”. In this paper, we excluded those whose residence was “outside of Pennsylvania” and included those residing in Washington, Greene, Beaver, Butler, Allegheny, Bedford, Fayette, and Westmoreland counties which resulted in a larger sample size.

See lines 87-100

3. What is the distribution of years that the participants took the survey in the present study? No emissions data was available in 2017, so did these subjects receive estimates from 2016? Participants were distributed among years, with 23, 9, 29, 20, and 17 in the respective years between 2012-2016. The 2016 emissions data were used for the 6 participants from 2017.

See lines 119-120 & 179-180

4. Switched from 2 to 5km buffer between the two studies, what was the rationale? In our first submitted manuscript, we included text in the methods “We applied GLM analyses using three spatial scales of cumulative well density: 1, 2, and 5 km. AIC criterion was used to determine the appropriate spatial scale to study” and in the results “Initial GLMs for cumulative well density at the three spatial scales against symptom total, including demographic variables, showed that models using cumulative well density in 5 km had the lowest AIC value and was therefore used as the defined radius in cumulative well density as an exposure variable (1 km: AIC=1095.26, 2 km: AIC=1039.73, 5 km: AIC=1024.86).” 

We have included this text back into the manuscript to explain why 5 km was chosen. 

See lines 216-218 & 254-256

1. Please add some explanation regarding the decision to combine 80+ (in many cases completely unrelated) symptoms into a single symptom count variable for the regression analysis. The health assessment was conducted using a standard clinical interview, which included a comprehensive list of symptoms (such as one might see at any clinical visit with a health care provider). We have edited lines 81-94 to more accurately reflect the process. The number of symptoms reported ranged from 0-36, as stated in the results on line 249-250. Other studies have used this approach—i.e., counting total number of symptoms reported as an indication of health (Rabinowitz for example). It is important to note that it is not unreasonable to expect emissions of several compounds, each with specific health effects, might cause multiple body systems to be affected. 

2. How does the present study compare to Brown et al. 2019 where only respiratory symptoms were assessed and no association was found with air emissions? Please add some text to the discussion section on this topic. This was added to the text: 

Brown et al. 2019 did not find an association with the median air emissions. The most likely explanation for this inconsistency is that for this study, we used the 90th percentile, rather than the median. 

See lines 336-338

3. The figures continue to be too low resolution to really read. Please update these to at least 300 dpi. We have used the PACE tool to provide the figures that meet journal requirements. Please download them and view on your computer to see if that enhances the quality compared to what is shown in the PDF. 

4. Add n (%) to table 1, please.

 Column added to table. 

5. Discussion, page 21, line 429+: the CWD metric is most strongly associated with total symptom count. To me, this indicates a physical and psychological pathway between UNGD and health. Some discussion of how perceived environmental change is associated with health might be helpful here. For example, the idea of solastalgia, feeling homesick even at home due to changed environmental conditions around the home (Albrecht). Further, in Lai 2017 “Understanding the psychological impact of unconventional gas developments in affected communities,” the authors find that negative perceptions of unconventional development was associated with negative psychological states. Thank you for this suggestion, we have edited the text

See lines 371-375

6. It should be emphasized that this study provides evidence that among people aware and concerned about UNGD the CWD metric performs best, we have no idea if this would be the case with people not concerned or pro-UNGD. There is an interplay of perception, psychology, and health that should be more carefully discussed. While it is true that our sample was aware of/concerned for their health, they did not know about the exposure metrics our study explored nor that their symptoms would be looked at in this manner. It is safe to assume that the bias in our sample is minimized by that fact. Our exposure metrics were determined and designed years later. We assume respondents could not have over or under reported symptoms related to the exposure metrics given they did not know them. Additionally, we do not know if people who are concerned vs. not concerned (supportive vs unsupportive) are different from each other since we did not test our metrics with individuals who were not concerned about UOGD.

See lines 84-85, 245, & 389-399

Reviewer 2

Line 39 - remove “of” before “50%” Addressed

Line 40 - change “grouping” to “groupings” Addressed

Line 51 - remove hyphen in “human-health” Addressed

Lines 62-63 - think you mean “Which exposure measure(s) is the best predictor…” or “Which exposure measures are the best predictors…” Addressed

Line 76 - did you mean “different” instead of “difference”? Addressed

MAJOR COMMENT: Lines 98-99 - it’s not clear why your analysis was restricted to the eight counties Analysis was restricted to PA residents (who happened to be from just 8 counties) because we used PA DEP emission data.

See lines 96-100

Line 116 - change “were plotted” to “was plotted” Addressed

Line 119 - remove “was completed” Addressed

Line 136 - change “rate” to “rates” Addressed

Line 143 - you mean “concentrations” not “emissions” Addressed

Lines 146, 147, 151 - “down-wind” “downwind” - choose one Addressed

MAJOR COMMENT: Line 156 - I still don’t understand what part wind direction played in your modeling The box model does not take wind direction into account. Brown et al. 2019 examined wind direction data from NOAA and found that the wind blows from the north, south, and west 90% of the time, with 10% coming from the east.

The use of putting sources in cardinal directions from a home was a broad way to think about wind direction but did not actually use the NOAA wind direction data in the analysis. This was updated in the text. 

See lines 148-180

MAJOR COMMENT: Line 159 - I still think you don’t adequately address in the paper the representativeness of the data at the Pittsburgh airport to the region at large. I don’t think it’s the only hourly station in the eight-county area that reports the variables you use. Why did you use this one station? The approach we took was to choose one major airport where we felt the most complete set of weather data could be gathered and generalized over an area. Data from smaller airports is frequently incomplete even if they are closer to a home. We recognize this is a limitation of this exploratory study.

See lines 162-168 & 416-417

Line 162 - change “was” to “were” Addressed

Line 163 - remove “yearly” as it implies annual-average weather, when in fact you used hourly weather data Addressed

Line 172 - change “respondents’” to “respondent’s” and “was” to “were” Addressed

MAJOR COMMENT: Paragraph beginning Line 173 - Your methodology treats the trajectory of each well’s plume equally when summing the quadrant concentrations together. In fact, a wind rose analysis would tell you how often winds blew from each quadrant toward each residence, allowing you to weight the concentrations from some quadrants more than others. You should either address this limitation or correct for it. We have addressed this in the limitations section.

See lines 411-414 

Line 173 - suggest changing “surround” to “are ubiquitous around” Addressed

Line 176 - I think you mean “in which quadrant the well was located relative to the residence”? Addressed

Line 182 - change “of their survey and was used” to “of their survey, which was used” Addressed

Line 186 - should “compared” be changed to “made”? Addressed

Line 201 - should “to generation” be “for determining”? Addressed

Line 213 - “a species” is singular but “taxa” is plural - change “taxa” to “taxon”? Addressed

Line 214 - change “are” to “is” Addressed

Line 223 - change “uses” to “used” Addressed

Line 232 - change “z score” to “z-score” Addressed

Line 248 - change “measures” to “measure” Addressed

I don’t think Figure 2 is cited in the text? Addressed

Lines 306, 314 - remove hyphen in “human-health” Addressed

MAJOR COMMENT: Line 311 - you only evaluated wells within 5 km of a residence, so how did you detect effects beyond that? We are stating that our study used a further range than references like [15,19] and detected effects beyond where those researchers studied. We did not look beyond 5 km. The text has been modified to help clarify that statement. 

See lines 318-321

Line 333 - remove “primarily”, all respondents were concerned about UOGD exposure Addressed

Line 342 - end of line, change “was” to “were” Addressed

Line 343 - remove hyphen in “exposure-magnitude” Addressed

Line 395 - I believe development and production are the ONLY UOGD stages, and I believe the literature supports that emissions from development tend to be higher than those from production Addressed

MAJOR COMMENT: Line 401 - your model actually does account for day-to-day weather patterns, as it utilizes hourly weather data. Do you mean that your model does not correct for differences in weather observations at residences relative to the weather station site? Yes, that is correct, the model does account for day to day weather patterns each day of the year. We have removed that line from the text. The main limitation is that the box model does not correct for topography. 

Line 406 - I think you mean “emissions concentrations”. Air dispersion modeling does not quantify emissions, it uses them to quantify concentrations. Also, your study is not the first to use dispersion modeling for UOGD purposes. Addressed

Line 411 - remove hyphen in “human-health” Addressed

Lines 417-418 - I think you want to remove “in and around communities at” Addressed

Line 419 - remove hyphen in “residence-level” Addressed

Line 420 - remove hyphen in “human-health” Addressed 

Line 577 - change “CDW” to “CWD” Addressed

---

## [Decision Letter · Decision Letter 2]

16 Jun 2020

PONE-D-19-34629R2

Exposure assessment of adults living near unconventional oil and natural gas development and reported health symptoms in southwest Pennsylvania, USA

PLOS ONE

Dear Dr. Blinn,

Thank you for submitting your manuscript to PLOS ONE. After careful consideration, we feel that it has merit but does not fully meet PLOS ONE’s publication criteria as it currently stands. Therefore, we invite you to submit a revised version of the manuscript that addresses the points raised during the review process.

We look forward to receiving your revised manuscript.

Kind regards,

Min Huang

Academic Editor

PLOS ONE

Additional Editor Comments (if provided):

Thanks for preparing and submitting the revised manuscript. One of the reviewers has some remaining comments which must be addressed before the manuscript can be published.

Reviewers' comments:

Reviewer's Responses to Questions

**Comments to the Author**

1. If the authors have adequately addressed your comments raised in a previous round of review and you feel that this manuscript is now acceptable for publication, you may indicate that here to bypass the “Comments to the Author” section, enter your conflict of interest statement in the “Confidential to Editor” section, and submit your "Accept" recommendation.

Reviewer #2: All comments have been addressed

Reviewer #3: (No Response)

2. Is the manuscript technically sound, and do the data support the conclusions?

Reviewer #2: Yes

Reviewer #3: Partly

3. Has the statistical analysis been performed appropriately and rigorously? 

Reviewer #2: Yes

Reviewer #3: Yes

4. Have the authors made all data underlying the findings in their manuscript fully available?

Reviewer #2: No

Reviewer #3: Yes

5. Is the manuscript presented in an intelligible fashion and written in standard English?

Reviewer #2: Yes

Reviewer #3: Yes

6. Review Comments to the Author

Reviewer #2: (No Response)

Reviewer #3: Please see my attached comments. The new revelation that 2016 well and weather data were matched to 2017 health data is a concern. Within my comments, you'll find a discussion on this. I will not recommend this for publication without addressing this issue. All other comments are minor/typographical.

7. PLOS authors have the option to publish the peer review history of their article (what does this mean?). If published, this will include your full peer review and any attached files.

Reviewer #2: No

Reviewer #3: Yes: Chris Holder

---

## [Author Response · Author response to Decision Letter 2]

17 Jul 2020

Comments Response

Reviewer 3

Line 51: change “adverse health” to “adverse health effects” (or outcomes, etc.) Addressed

Line 85: data is plural, so change “was abstracted” to “were abstracted” Addressed

Lines 95–96: convert “General” to lower case; suggest using semicolons in the list so that the EENT group is separated more clearly from skin; also, need to put the EENT group in the right order. So: “general; lung and heart; skin; eyes, ears, nose, and throat; gastrointestinal (GI); nerves and muscle; reproductive; blood system; and psychological” Addressed

Line 99: 14 individuals were excluded because you didn’t know their home lat/longs right? If so, move the parenthetical to the end of the previous sentence Addressed

Lines 105–106: If no respondents lived in Lawrence County, then why color it dark gray indicating “county with study residencies”? (did you mean “residences”?) Why does it matter that someone in Butler County lived near the Lawrence border? Lawrence County was colored dark grey because we included gas wells from the county and considered the wells part of the study. The individual living in Butler County lived near the border of Lawrence and a well pad located within their 5-km radius was located in Lawrence, so we had to include those wells in the study. For sake of not identifying the respondent, all Lawrence County wells were included in the analysis. 

MAJOR COMMENT Lines 119–120 (and 168): You used 2016 well and weather data for comparison to 2017 health survey data? How is that appropriate? Your data are already of low temporal resolution (one-time reporting of symptoms related to annual well data), and with this mismatching of 2016/2017, I find myself wondering how much different your study outcomes would be if you just randomized which year of data were applied to each respondent. You need a discussion of this in your discussion/limitations, or remove 2017 from your assessment, or redo the 2017 assessment with 2017 well/weather data if available We were able to use 2017 weather and gas well data for the 2017 health data and updated our models and findings accordingly. Now, each survey year has corresponding weather and gas well data from the same year. Each exposure metric remained significant, while the Titan results changed slightly. New figures have been uploaded. 

Lines 148–149: remove either “were derived” or “were estimated”, it’s redundant Addressed

Line 150: I think you should use “respondent’s” instead of “participant’s” for consistency. Addressed

Line 165: “hourly conditions for each hour” is redundant Addressed

Line 208: change “determining” to “determine” Addressed

Line 255: change “and was therefore selected” to “and were therefore selected” Addressed

Line 316: alter to “Despite a high degree of inherent complexity in associations between health and UOGD…” Addressed

Line 328: change “attempt” to “attempts” Addressed

Lines 354–356: this is an incomplete sentence Addressed

Line 397: I think you should use “respondents” instead of “participants” for consistency. Addressed – also addressed in S2 Appendix

---

## [Editor Report · Decision Letter 3]

27 Jul 2020

Exposure assessment of adults living near unconventional oil and natural gas development and reported health symptoms in southwest Pennsylvania, USA

PONE-D-19-34629R3

Dear Dr. Blinn,

We’re pleased to inform you that your manuscript has been judged scientifically suitable for publication and will be formally accepted for publication once it meets all outstanding technical requirements.

Kind regards,

Min Huang

Academic Editor

PLOS ONE

Additional Editor Comments (optional):

Thanks to the authors and the reviewers for their efforts. In my view all reviewers' comments have been addressed.
---

## [Editor Report · Acceptance letter]

5 Aug 2020

PONE-D-19-34629R3 

Exposure assessment of adults living near unconventional oil and natural gas development and reported health symptoms in southwest Pennsylvania, USA 

Dear Dr. Blinn:

I'm pleased to inform you that your manuscript has been deemed suitable for publication in PLOS ONE. Congratulations! Your manuscript is now with our production department. 

Kind regards, 

on behalf of

Dr. Min Huang 

Academic Editor

PLOS ONE